# Structural Elucidation of Antibiotic TKR2999, an Antifungal Lipodepsipeptide Isolated from the Fungus *Foliophoma fallens*

**DOI:** 10.3390/antibiotics9060278

**Published:** 2020-05-26

**Authors:** Gloria Crespo, Ignacio Pérez-Victoria, Francisco Javier Ortiz-López, Víctor González-Menéndez, Mercedes de la Cruz, Bastien Cautain, Pilar Sánchez, Francisca Vicente, Olga Genilloud, Fernando Reyes

**Affiliations:** Fundación MEDINA, Centro de Excelencia en Investigación de Medicamentos Innovadores en Andalucía, Avenida del Conocimiento 34, Parque Tecnológico de Ciencias de la Salud, E-18016 Granada, Spain; gloria.crespo@medinaandalucia.es (G.C.); ignacio.perez-victoria@medinaandalucia.es (I.P.-V.); javier.ortiz@medinaandalucia.es (F.J.O.-L.); victor.gonzalez@medinaandalucia.es (V.G.-M.); mercedes.delacruz@medinaandalucia.es (M.d.l.C.); cautainbastien@gmail.com (B.C.); pilar.sanchez@medinaandalucia.es (P.S.); francisca.vicente@medinaandalucia.es (F.V.); olga.genilloud@medinaandalucia.es (O.G.)

**Keywords:** *Foliophoma fallens*, antibiotic TKR2999, lipodepsipeptide, structural elucidation, Marfey’s analysis, antifungal activity

## Abstract

An antifungal lipodepsipeptide was obtained from cultures of the fungus *Foliophoma fallens* CF-236885. Its structure, elucidated by HRMS and NMR spectroscopy, contained Gly, Thr, Asn, β-Ala, Orn, Ala, two Ser residues, and 3-hydroxy-4-methylhexadecanoic acid. The absolute configuration of its amino acid residues was determined using Marfey’s analysis and *J*-based configuration analysis helped to establish the relative configuration of the 3-hydroxy-4-methylhexadecanoic acid moiety. A literature search retrieved a patent describing antibiotic TKR2999 (**1**), whose non-disclosed structure was confirmed to be identical to that found for our compound, according to its physicochemical properties and NMR spectra. Compound **1** displayed potent antifungal activity against *Aspergillus fumigatus* and a panel of *Candida* strains.

## 1. Introduction

Invasive fungal infections are still a major health problem, especially when affecting immunocompromised patients [1]. *Candida* species and *Aspergillus fumigatus* remain the main pathogens responsible for such infections. Current treatments include the use of amphotericin B, triazoles such as ketoconazole or fluconazole, and echinocandin-like lipopeptides, represented by three molecules in clinical use, namely caspofungin, micafungin, and anidulafungin [2,3,4]. Resistant strains are emerging, and the existence of new alternatives to enhance the efficacy or even replace these treatments are urgently required.

Natural products offer an interesting source of new molecules and fungal metabolites have proven to be a rich source of novel agents with potential against fungal infections. A survey of a subset of 8320 microbial acetone extracts from our proprietary collection against five *Candida* opportunistic strains (*Candida glabrata*, *C. krusei*, *C. parapsilosis*, *C. tropicalis*, *C. albicans*) and *Aspergillus fumigatus* identified, after liquid chromatography-mass spectrometry (LC/MS) dereplication, ten bioactive extracts with promising chemical profiles [5]. One of these extracts, obtained after cultivation of the fungus *Foliophoma fallens* in CMK medium, displayed a liquid chromatography-ultraviolet (LC-UV) profile containing almost a unique peak (Figure 1) whose HRMS data suggested a peptidic nature for the molecule. Based on the coincident physicochemical properties and NMR spectra, the compound was identified as antibiotic TKR2999 (**1**), an antifungal peptide previously described in a patent whose structure was not disclosed in the document [6]. A number of natural peptide-like compounds have been identified in the past as potent antifungal agents, and examples include the above-mentioned echinocandins, phaeofungin, a lipodepsipeptide isolated from a *Phaeosphaeria* sp. using the *C. albicans* fitness test [7], or phomafungin, isolated from a *Phoma* sp. using the same approach [8]. The latter two compounds share a strong structural resemblance with compound **1** and phomafungin even shares the same 3-hydroxy-4-methylhexadecanoic acid structural unit. 

*Foliophoma* was recently introduced as a new camarosporium-like fungal genus, named after its association with leaf spots, and its morphological similarity to *Phoma* [9]. It includes only two species, *F. fallens* (type species) [9] and the recently introduced *F. camporesii* [10]. No reports on the chemical composition of these two species or their synonymic species *Phoma fallens*, *Phoma glaucispora*, *Pleospora fallens*, *Phyllosticta glaucispora*, and *Phyllosticta oleandri* have been published so far.

## 2. Results and Discussion

To accomplish the full structural elucidation of compound **1**, a 1 L culture of *F. fallens* in CMK medium was extracted with acetone and, after centrifugation, decantation, and removal of the acetone under a nitrogen stream, a precipitate was formed. Antifungal activity was detected in this solid precipitate, which was filtered, dissolved in DMSO and purified by preparative HPLC to yield pure compound **1**. Its molecular formula was established as C_44_H_78_N_10_O_14_ based on the existence of a protonated ion at m/z 971.5797 in its ESI-TOF analysis and the presence of 44 signals in its ^13^C NMR spectrum. Inspection of the NMR spectra acquired in DMSO-*d*_6_ (Table 1) immediately evidenced the peptidic nature of the molecule, with signals in the region between 7.0 and 8.5 ppm of its ^1^H NMR spectrum accounting for the presence of amide NH protons; signals between 3.5 and 5.0 ppm accounting for α-amino acid protons; and 10 signals in the low field region of its ^13^C NMR spectrum (169–173 ppm) accounting for the presence of amide or ester carbonyl groups. Extensive analysis of 2D NMR spectra (COSY, HSQC, HMBC, and HSQC-TOCSY) allowed the identification in the molecule of the following amino acid residues: Gly, Thr, Asn, 2 × Ser, β-Ala, Orn, and Ala, together with a lipidic moiety whose identity was established as 3-hydroxy-4-methylhexadecanoic acid (HMHDA) (Figure 2). Signals for a methylene group at δ_H_ 2.31 and 2.39 (H-2) that correlated in the COSY spectrum with an oxygenated methine at δ_H_ 5.04 ppm (H-3), which in turn correlated with a methine at δ_H_ 1.68 ppm (H-4) coupled to the doublet methyl group at δ_H_ 0.82 ppm (4-Me) and a methylene at δ_H_ 1.31 and 1.04 ppm (H-5), and the existence of three intense HMBC correlations from the latter methyl group to carbons at δ_C_ 75.2 (C-3), 35.8 (C-4), and 31.6 (C-5) unequivocally confirmed the presence of this HMHDA unit in the molecule. The sequencing of all the residues was performed using a combination of HMBC and NOESY data. 

Key long-range correlations from α-protons to carbonyl carbons of adjacent amino acids plus NOESY correlations between α-protons and NH protons of adjacent amino acids allowed this sequence to be established as Gly-Thr-Asn-Ser1-Ser2-β-Ala-Orn-Ala-HMHDA (Figure 3). Additionally, a ring closure via an ester bond between the carbonyl group of Gly and the 3-hydroxy group of HMHDA was confirmed by the existence of an HMBC correlation between the H-3 proton of HMHDA at δ_H_ 5.04 ppm and the carbonyl carbon of Gly at δ_C_ 169.0 ppm. 

Once the planar structure of the molecule had been established, Marfey’s analysis [11] including UV and LC/MS detection was applied to determine the absolute configuration of all the chiral amino acid residues. Hydrolysis of the peptide overnight at 110 °C and derivatization of the hydrolyzate with L-FDVA led to the identification of D-*allo*-Thr, D-Asn, L-Ser (× 2), L-Orn, and L-Ala as the chiral amino acid components of antibiotic TKR2999.

To determine the absolute configuration of the two chiral centers present in the 3-hydroxy-4-methylhexadecanoic acid moiety, a strategy including *J*-based configuration analysis (JBCA) [12] in combination with derivatization with Mosher reagents [13] of the 3-OH group of the molecule obtained after hydrolysis of the ester functionality was attempted. However, reaction of the hydrolyzed lipopeptide with the Mosher reagent was unsuccessful and attempts to obtain the 3-hydroxy-4-methylhexadecanoic acid moiety forcing the hydrolysis in acidic conditions yielded the dehydration product. Nevertheless, it was possible to determine the relative configuration of the contiguous chiral centers, C-3 and C-4, in the fatty acid moiety by means of JBCA and NOESY/ROESY analyses. The key ^3^*J*_HH_ for the geminal protons H-2a and H-2b, and proton H-3 at the esterification position of the HMHDA moiety were determined from both the ^1^H NMR and *J*-resolved spectra acquired at 50 °C (see Appendix A), since at this higher temperature not only were the signals sharpened, but were also better resolved, facilitating the direct measurement of the key couplings. On the other hand, the key ^3^*J*_HC_ between H-3 and the carbon of the methyl substituent at C-4 was derived from the *J*-HMBC experiment (see Appendix A) [14]. These key coupling constants were interpreted based on their Karplus-type dependence on dihedral angles [15,16] and were also qualitatively classified as small, medium, or large as proposed for the JBCA method to distinguish between a possible dominant conformer or rather an equilibrium of rotamers. In this manner, H-3 was found to be approximately antiperiplanar to H-2a (ϕ ca. 145° derived from ^3^*J*_HH_ = 9.3 Hz) displaying a dihedral angle ϕ ca. −68° (derived from ^3^*J*_HH_ = 3.0 Hz) with H-2b. A NOESY/ROESY correlation was also observed between H-3 and H-2b. On the other hand, the 5.3 Hz coupling constant between H-3 and H-4 can be considered to be of medium magnitude compared to the expected ranges for *anti* (8-11 Hz, large) or *gauche* (1–4 Hz, small) arrangements in monooxygenated 1,2-methine systems [12], indicating the contribution of more than one rotamer along the C-3 to C-4 bond. The long-range coupling ^3^*J*_CH_ = 4.3 Hz between H-3 and the methyl was also of medium magnitude compared to the expected ranges for *anti* (6–8 Hz, large) or *gauche* (1–3 Hz, small) displays, further confirming an equilibrium of rotamers [12]. According to the JBCA approach these medium magnitude coupling constants mentioned are compatible with the A-1/A-3 pair of rotamers (Figure 4) for a *threo* relative stereochemistry, or with the B-1/B-3 for the *erythro* diastereomer (Appendix A) The equilibrium between the indicated two pairs of rotamers was further confirmed by the key NOESY/ROESY correlations observed between H-4/H-2, H-4/H-3, 4-Me/H-2a, and 4-Me/H-3 (Figure 4). Interestingly, the A-1 rotamer of the *threo* configuration was found in the crystal structure of the lipodepsipeptide oryzamide A, which contains the homologous fatty acid 3-hydroxy-4-methyldecanoic acid (HMDA) [17], while the A-3 rotamer was found in the crystal structure of lipodepsipeptide scopularide A, which contains the homologous 3-hydroxy-4-methylhexanoic acid (HMHA) [18]. The occurrence of both rotamers in the solid state also supports their equilibrium in solution. To finally discriminate which rotamer pair corresponds to **1**, and evaluation of the long-range coupling ^3^*J*_CH_ between H-4 and C-2 provided the answer. A quantitative measurement was not possible due to the absence of the corresponding cross-peak in the J-HMBC experiment. However, such cross-peak absence also occurred in the standard HMBC spectrum acquired at an optimized value of 8 Hz, clearly indicating that the long-range coupling involved must be small, a result only compatible with the A-1/A-3 rotamer pair for the *threo* relative configuration. Figure 4 summarizes the application of the JBCA approach and NOESY/ROESY analyses to determine the relative 3*S**, 4*S** configuration for the two chiral centers of the fatty acid.

Unfortunately, it was not possible to correlate the relative configuration of the fatty acid moiety of TKR2999 with the absolute configuration of its constituent amino acids. Nevertheless, based on its fungal origin, we tentatively propose that the HMDA moiety in oryzamides from *Nigrospora oryzae* [17] and scopularides from *Scopulariopsis* sp. CMB-F458 [18], the homologous 3-hydroxy-4-methylhexanoic acid (HMHA) moiety in chrysogenamide D from *Penicillium chrysogenum* [19], and the homologous HMHDA unit of TKR2999 share all the same 3*S*, 4*S* absolute configuration for the two chiral centers of the fatty acid chain. 

The antifungal activity of compound **1** was tested against a panel of five yeasts and one fungal strain including five *Candida* species, namely *C. glabrata*, *C. krusei*, *C. parapsilosis*, *C. tropicalis*, and *C. albicans*, and one strain of *A. fumigatus*. Potent bioactivity with minimum inhibition concentration (MIC) values ranging from 0.5 to 2 μg/mL against all the pathogenic strains were obtained, while **1** displayed moderate toxicity against the THLE-2 epithelial human cell line (Table 2). 

In conclusion, the structure including the full absolute configuration of most of the structural units of the fungal depsipeptide antibiotic TKR2999 was determined and the antifungal activity previously reported [6] was confirmed in an extended panel of five *Candida* and one *Aspergillus* species. This report constitutes the first chemical study of fungi in the genus *Foliophoma* and once again confirms the importance of fungal natural products as a valuable source of antifungal compounds [20].

## 3. Materials and Methods 

### 3.1. General Experimental Procedures

Optical rotations were measured on a Jasco P-2000 polarimeter (JASCO Corporation, Tokyo, Japan). IR spectra were recorded with a JASCO FT/IR-4100 spectrometer (JASCO Corporation) equipped with a PIKE MIRacle^TM^ single attenuated total reflection (ATR) accessory. 1D and 2D NMR spectra were recorded on a Bruker Avance III spectrometer at 500/125 MHz (^1^H/^13^C NMR, respectively) equipped with a 1.7 mm TCI MicroCryoProbe^TM^ (Bruker Biospin, Fällanden, Switzerland). Chemical shifts were reported in ppm using residual DMSO-*d*_6_ signals (δ 2.51 for ^1^H and 39.0 for 13C) as the internal reference. HMBC experiments were optimized for a ^3^*J*_CH_ of 8 Hz. ESI-TOF spectra were acquired using a Bruker maXis QTOF (Bruker Daltonik GmbH, Bremen, Germany) mass spectrometer coupled to an Agilent 1260 RR HPLC (Agilent Technologies, Waldbronn, Germany). Preparative HPLC separations were performed on a Gilson GX-281 322H2 instrument (Gilson Technologies, USA).

### 3.2. Producing Fungus and Its Characterization

The producer microorganism (CF-236885) was isolated from *Pinus* sp. collected in Lisbon (Portugal) using a dilution to extinction method in 48-well plates containing YMC medium [21]. Frozen stock cultures in 10% glycerol (−80 °C) are maintained in the fungal collection of Fundación MEDINA. Furthermore, to estimate the approximate phylogenetic position of strain CF-236885, genomic DNA was extracted from mycelia grown on malt-yeast extract agar. DNA extraction, PCR amplification, and DNA sequencing were performed as previously described [22]. Sequences of the complete ITS1-5.8S-ITS2 and initial 28S region or independent ITS and partial 28S rDNA sequences were compared with those deposited at GenBank or the National Institute of Technology and Evaluation (NITE) Biological Resource Center [23] by using the BLAST application. Database matching with the ITS rDNA sequence [24] yielded a complete sequence similarity (100%) to the type strain of *Foliophoma fallens* CBS 284.70, thus indicating that strain CF-236885 was genetically similar to *F. fallens*, and conspecific. High similar scores to other validated fungal strains of this species (e.g., *F. fallens* CBS 161.78 (99.426%)), indicated that CF-236885 can be classified as *Foliophoma fallens* (Coniothyriaceae, Pleosporales).

### 3.3. Fermentation

Strain CF-236885 (*F. fallens*) was fermented by inoculating ten mycelial agar plugs into SMYA medium (Bacto neopeptone 10 g; maltose 40 g; yeast extract 10 g; agar 3 g; H_2_O 1 L) in a flask (50 mL medium in a 250 mL Erlenmeyer). The flask was incubated on a rotary shaker at 220 rpm at 22 °C with 80% relative humidity. After growing the seed stage for seven days, a 1.5 mL aliquot was used to inoculate each flask of the production medium CMK (D-cellobiose 40 g; yeast extract 1 g; glycerol 2 g; Murashige and Skoog salts (SIGMA M-5524) 4.3 g, and distilled water 1 L). The 20 flasks (50 mL medium per 250 mL unbaffled flask) were incubated at 22 °C on a rotary shaker (220 rpm, 5 cm throw) for 23 days, 80% relative humidity.

### 3.4. Extraction and Isolation

A 1-L culture of CF-236885 was extracted by the addition of acetone (1 L), agitation at 220 rpm for 1 h, centrifugation at 8500 rpm, filtration, and evaporation of the organic solvent under a nitrogen stream. The solid precipitate formed was dried to obtain 196 mg of a material that was dissolved in DMSO and subjected to repeated preparative HPLC injections (X-Bridge C18, 19 × 250 mm, 5 μM, gradient H_2_O + 0.1% TFA-CH_3_CN + 0.1% TFA from 35% to 60% organic in 25 min, 14 mL/min, UV detection at 210 and 280 nm) to yield 18.9 mg of compound **1**. 

Antibiotic TKR2999 (**1**): white, amorphous solid; [α]_20_^D^ −15.1 (c 0.08, MeOH); IR (ATR) ν_max_: 3304, 2922, 2852, 2831, 1744, 1678, 1630, 1536, 1435, 1406, 1313, 1202, 1012, 951 cm^−1^; (+)-ESI-TOFMS *m/z* 971.5797 [M+H]^+^ (calcd for C_44_H_79_N_10_O_14_^+^, 971.5772); ^1^H and ^13^C NMR data see Table 1.

### 3.5. Marfey’s Analysis of Compound 1

A sample (0.5 mg) of compound **1** was dissolved in 1 mL of 6N HCl and heated at 110 °C for 16 h. After evaporation to dryness under a nitrogen stream, the hydrolyzate was dissolved in 100 µL of water. A 1% (w/v) solution (100 μL) of L-FDVA (Marfey’s reagent, N-(2,4-dinitro-5-fluorophenyl)-L-valinamide) in acetone was added to the aqueous solution of the peptide hydrolyzate or to an aliquot (50 μL) of a 50 mM solution of each amino acid standard (D or L). After the addition of 20 μL of 1 M NaHCO_3_ solution, each mixture was incubated at 40 °C for 60 min. The reactions were quenched by the addition of 10 μL of 1N HCl and the mixtures were diluted with 700 μL of acetonitrile and analyzed by LC/MS on an Agilent 1100 single Quadrupole LC/MS instrument using a Waters X-Bridge C18 column (4.6 × 150 mm, 5 μm) maintained at 40 °C. A mixture of two solvents, A (10% acetronitrile, 90% water) and B (90% acetronitrile, 10% water), both containing 1.3 mM trifluoroacetic acid and 1.3 mM ammonium formate, was used as the mobile phase under a linear gradient elution mode (10 to 30% B in 35 min, 30 to 100% B in 1 min, isocratic 100% B for 4 min) at a flow rate of 1 mL/min. Retention times (min) for the derivatized (L-FDVA) amino acid standards under the reported conditions were as follows: Gly: 16.43; L-Thr: 12.71; D-Thr: 18.47; L-*allo*-Thr: 13.20; D-*allo*-Thr: 15.80; L-Asp: 13.20; D-Asp: 14.84; L-Ser: 12.42; D-Ser: 13.88; β-Ala: 19.46; L-Orn (monoderivatized): 10.10; D-Orn (monoderivatized): 9.31; L-Orn (diderivatized): 30.35; D-Orn (diderivatized): 28.95; L-Ala 16.37; D-Ala: 21.52. Retention times (min) for the observed peaks in the HPLC trace of the L-FDVA derivatized hydrolysis product of compound 1 were: Gly: 16.38; D-*allo*-Thr: 15.82; D-Asp: 14.93; L-Ser: 12.45; β-Ala: 19.49; L-Orn (monoderivatized): 10.17; L-Orn (diderivatized): 30.39; L-Ala 16.38.

### 3.6. Biological Activity

Antifungal activity against *C. albicans* ATCC64124, *C. glabrata* ATCC2001, *C. krusei* ATCC6258, *C. parapsilosis* ATCC22019, *C. tropicalis* ATCC750, and *A. fumigatus* ATCC 46645, and cytotoxicity against the THLE-2 (ATCC CRL-10149) epithelial human cell line were performed as previously described [5,25]. 

## Figures and Tables

**Figure 1 antibiotics-09-00278-f001:**
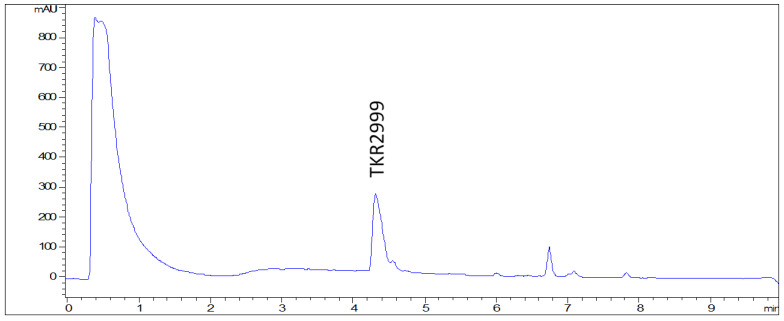
LC-UV (210 nm) chromatogram of the extract containing compound **1**.

**Figure 2 antibiotics-09-00278-f002:**
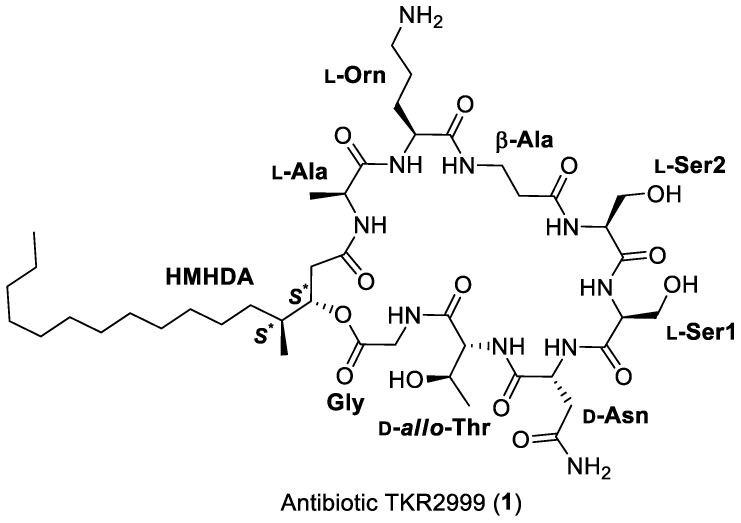
Structure of antibiotic TKR 2999 (1).

**Figure 3 antibiotics-09-00278-f003:**
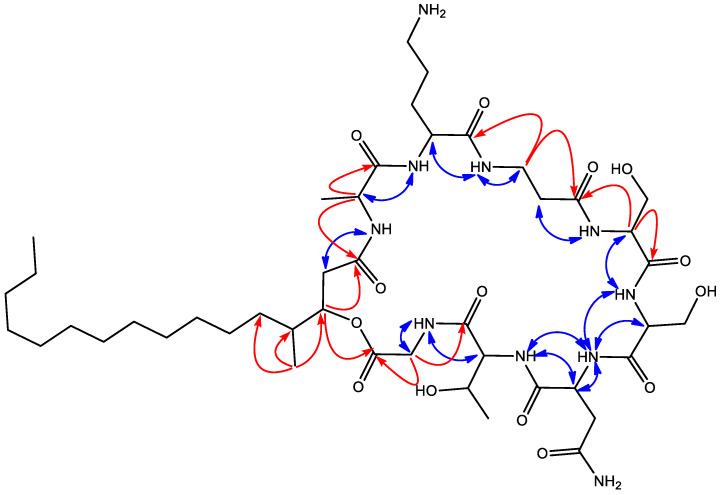
Key NOESY (blue) and HMBC (H to C, red) correlations observed in the spectra of compound **1**.

**Figure 4 antibiotics-09-00278-f004:**
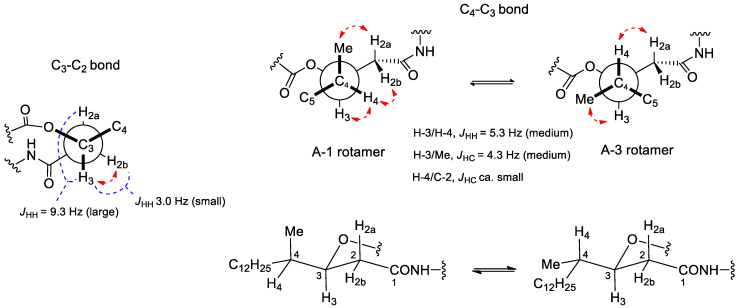
JBCA approach and key NOESY/ROESY correlations (dashed red arrows) employed to determine the *threo* (3*S**, 4*S**) relative stereochemistry for the two chiral centers of the HMHDA moiety in compound **1**.

**Table 1 antibiotics-09-00278-t001:** NMR data of antibiotic TKR2999 (**1**) in DMSO-*d*_6_.

Position	δ_C_	δ_H_, Mult (*J* in Hz)	Position	δ_C_	δ_H_, Mult (*J* in Hz)
**Gly**			**Orn**		
1	169.0		1	170.9	
2	40.9	3.98 dd (17.4, 4.6) 3.72 dd (17.4, 6.3)	2	52.0	4.20 m
NH		8.16 t (5.3)	3	28.5	1.77 m 1.52 m
**Thr**			4	23.6	1.52 m
1	170.2		5	38.6	2.75 m
2	58.3	4.24 m	NH		7.99 m
3	66.8	3.95 m	NH_2_		7.63 m
4	19.9	1.07 d (6.2)	**Ala**		
NH		7.67 d (8.7)	1	172.5	
3-OH		4.87 d (4.3)	2	49.0	4.17 m
**Asn**			3	17.9	1.23 m
1	170.6		NH		7.99 m
2	50.0	4.57 m	**HMHDA**		
3	36.9	2.65 dd (15.5, 5.7) 2.50 m	1	169.8	
4	172.1		2	36.7	2.39 dd (14.4, 9.3) ^a^ 2.31 dd (14.4, 2.7) ^a^
NH		8.32 d (8.0)	3	75.2	5.04 ddd (8.6, 5.3, 3.2) ^a^
NH2		7.43 s 7.01 s	4	35.8	1.68 m
**Ser1**			54	31.6	1.31 m 1.04 m
1	170.1		6	26.5	1.31 m 1.17 m
2	56.0	4.23 m	7	29.2	1.24 m
3	61.2	3.63 m 3.59 m	8	29.1	1.24 m
NH		7.99 m	9	29.1	1.24 m
3-OH		4.94 t (4.9)	10	29.1	1.24 m
**Ser2**			11	29.1	1.24 m
1	170.7		12	29.1	1.24 m
2	55.3	4.30 m	13	28.7	1.24 m
3	61.5	3.63 m 3.54 m	14	31.3	1.23 m
NH		8.03 d (7.6)	15	22.1	1.26 m
3-OH		5.04 m	16	14.0	0.85 t (6.8)
**β-Ala**			4-Me	14.7	0.82 d (6.6)
1	171.3				
2	35.1	2.36 m			
3	35.5	3.34 m 3.29 m			
NH		7.76 t (5.3)			

^a^ Coupling constants determined from the ^1^H and *J*-resolved spectra acquired at 50 °C.

**Table 2 antibiotics-09-00278-t002:** Antifungal (MIC, *Candida*, and *Aspergillus* strains) and cytotoxic (IC_50_, THLE-2 cell line) activity of compound **1**.

Fungal Strain/Cell Line
Compound	*C. glabrata*	*C. krusei*	*C. parapsilosis*	*C. tropicalis*	*C. albicans*	*A. fumigatus*	THLE-2
MIC (µg/mL)	IC_50_ (µg/mL)
**1**	1–0.5	2–1	2–1	1–0.5	1	1–0.5	5.38 ± 0.91
Amphotericin B	2	4	4	4	4	4

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
