# Peer review of "Structural Elucidation of Antibiotic TKR2999, an Antifungal Lipodepsipeptide Isolated from the Fungus Foliophoma fallens"

_antibiotics, 2020, doi:10.3390/antibiotics9060278_

Round 1

Reviewer 1 Report

The manuscript by Gloria Crespo et al investigates, via NMR, the bioactivity and the molecular structure of an antifungal lipodepsipeptide obtained from Foliophoma fallens CF-236885 cultures.

Actually, as the same authors recognize, the biological activity of this compound was already described in a patent (TKR2999). Thus, the only novelty of the presented study lies in the characterization of the molecular structure that was unknown to date.

Although the manuscript is well written, in my opinion, in consideration the obtained results, it could be more suitable for a short communication then as an full paper.

However, it is necessary that the authors make the small changes indicated below:

add statistical analysis to biological data

Enrich the introduction with some news on Foliophoma fallens

Author Response

We acknowledge the work of this reviewer and his/her comments to improve the quality of our article. Regarding the concerns expressed, we have implemented the following changes:

1) We recognize that the value of our article lies in the characterization and disclosure of the structure of the antifungal compound, unknown to date but please also note that this article also constitutes the first chemical report on the chemical composition of the recently introduced Foliophoma fungal genus.

2) Please note that short communications is not under the categories of manuscripts that can be sent to Antibiotics.

3) MIC values are usually expressed as the concentration of interval of concentrations tested that cause a total inhibition of the microbial strain. No statistical analysis is performed in this case. In the case of the THLE-2 cell line inhibition we have added the SD obtained after statistical treatment of the data

4) A paragraph with a brief description of the genus Foliophoma, its species and the lack of previous reports on the chemistry of these species has been added to the introduction of the article.

Reviewer 2 Report

This manuscript described the isolation and structural elucidation of a desipeptide from cultures of the fungus Foliophoma fallens as well as its antifungal activity. The relative configuration of the fatty acid moiety discussed in line 117 seems to be ambiguous. In this section, the 3JH3-H4 ( 5.3 Hz) is ascribed due to a dihedral angle of ca. 45 degree between H-3 and H-4; however, a dihedral angle of ca. 140 degree also have similar coupling value. Moreover, according to the JBCA approach, the 5.3 Hz (3JH3-H4) is considered as a medium value (7-10 Hz large; 0-3 Hz small), resulting from a mixture of anti and gauche conformers. In Fig. 4, only one conformer (C-3-C-4) is discussed in the text, and the 2JCH values are not discussed in the text, which could not clearly present the real conformers of this moiety.

In addition, there is some ambiguous descriptions about the configuration at C-3 and C-4 of the fatty acid residue. In lines 132-137, the author gave some example compounds with homologous fatty acid residues to HMHDA of compound 1, and suggested that those homologous residues have the same 3S,4S-configuration. Whereas, the fatty acid residue in Fig. 2 is drawn as a relative configuration (3S,4S). Is the configuration of this moiety relative or absolute?

There are some typo need to be revised and concerns need to be clarified. Those were listed as follows:

  1. lines 93-94: “...configuration of all….centers” revised to “…configurations of all the chiral amino acid residues.”
  2. line 95: “identified" revised to “led to the identification of ”
  3. line 96: “structural units”revised to “amino residues”
  4. H-2 of HMHDA, Table 1: 2a and 2b were not found. Only one proton resonance is shown.
  5. line 122: “…the relative stereochemistry 3S,4S …” revised to “the relative 3S,4S-configuration…”
  6. line 140: delete the comma between “bioactivity” and “with”.
  7. line 141-142: “… strains tested were obtained. Additionally, the compound displayed…” revised to “…strains were obtained, while 1 displayed…”
  8. line 205: “aliquots” revised to “an aliquot”

Author Response

This manuscript described the isolation and structural elucidation of a desipeptide from cultures of the fungus Foliophoma fallens as well as its antifungal activity. The relative configuration of the fatty acid moiety discussed in line 117 seems to be ambiguous. In this section, the 3JH3-H4 ( 5.3 Hz) is ascribed due to a dihedral angle of ca. 45 degree between H-3 and H-4; however, a dihedral angle of ca. 140 degree also have similar coupling value. Moreover, according to the JBCA approach, the 5.3 Hz (3JH3-H4) is considered as a medium value (7-10 Hz large; 0-3 Hz small), resulting from a mixture of anti and gauche conformers. In Fig. 4, only one conformer (C-3-C-4) is discussed in the text, and the 2JCH values are not discussed in the text, which could not clearly present the real conformers of this moiety.

We appreciate the work and the comments of the referee regarding this point. It is completely right that the experimental 3JH3-H4 = 5.3 Hz should be considered a medium value compared to the standard ranges for monooxygenated substitution patterns in 1,2-methines: 8-11 Hz large and 1-4 Hz small (the ranges provided by the reviewer actually corresponds to a deoxygenated substitution pattern, which is not the case for compound 1). Our proposal was bsed on the assumption that a predominant conformer was present in the equilibrium due to the large heteronuclear coupling constant between H-3 and 4-Me. Revisiting the J-HMBC experiment we have realized that we employed an erroneous scaling factor in the calculation. This error has been amended using now the correct scaling factor which provides a value of 4.3 Hz for the heteronuclear coupling constant between H-3 and 4-Me. This value is also a medium value, indicating an equilibrium of rotamers as pointed out by the reviewer. Furthermore, the pattern of NOESY/ROESY key correlations, where we had overlooked the important correlation between H-3 and the methyl protons, further supports the equilibrium between two pairs of rotamers, A-1/A-3 for the threo configuration and B-1/B-3 for the erythro configuration. To finally discriminate the actual pair of rotamers, we focused on the discriminating value of 3JH4-C2. Unfortunately, no crosspeak is observed in the J-HMBC spectrum due to poor signal to noise ratio. Nevertheless, the expected peak is also absent in the standard HMBC spectrum, acquired at an optimized J of 8Hz, clearly indicating that the coupling involved must be small and thus rendering A-1/A-3 as the actual rotamer pair, which correspond to the threo relative configuration. Interestingly the A-1 rotamer is found in the crystal structure of the lipodepsipeptide oryzamide A (which contains a homologous fatty acid), while the A-3 rotamer is observed in the crystal structure of scopularide A (another lipodepsipeptide containing an homologous fatty acid). The occurrence of both rotamers in the solid state also supports their equilibrium in solution.

The text has been modified to include this revised determination of the relative configuration and a new figure has been added to the supplementary information.

In addition, there is some ambiguous descriptions about the configuration at C-3 and C-4 of the fatty acid residue. In lines 132-137, the author gave some example compounds with homologous fatty acid residues to HMHDA of compound 1, and suggested that those homologous residues have the same 3S,4S-configuration. Whereas, the fatty acid residue in Fig. 2 is drawn as a relative configuration (3S,4S). Is the configuration of this moiety relative or absolute?

Please note that the configuration that can be determined using the J-based configurational analysis is relative. We postulate that the absolute configuration should be the same as on other similar compounds where a 3S,4S-configuration has always been found

There are some typo need to be revised and concerns need to be clarified. Those were listed as follows:

  1. lines 93-94: “...configuration of all….centers” revised to “…configurations of all the chiral amino acid residues.”
  2. line 95: “identified" revised to “led to the identification of ”
  3. line 96: “structural units”revised to “amino residues”
  4. H-2 of HMHDA, Table 1: 2a and 2b were not found. Only one proton resonance is shown.
  5. line 122: “…the relative stereochemistry 3S,4S …” revised to “the relative 3S,4S-configuration…”
  6. line 140: delete the comma between “bioactivity” and “with”.
  7. line 141-142: “… strains tested were obtained. Additionally, the compound displayed…” revised to “…strains were obtained, while 1 displayed…”
  8. line 205: “aliquots” revised to “an aliquot”

All these typos have been corrected.

Round 2

Reviewer 2 Report

The authors have made a suitable response to my comments and the manuscript has been revised accordingly. However, I found some additional typos that were listed as follows:

  1. line 177: “13C” revised to “13C”.
  2. line 180 “3JCH” revised to “3JCH”.
  3. line 202: a space between "250" and "mL" is needed.
  4. lines 224-225: “IR (ATR) v cm-1: 3304, 2922, 2852, 2831, 1744, 1678, 1630, 1536, 1435, 1406, 1313, 1202, 1012, 951” revised to “IR (ATR) vmax 3304, 2922, 2852, 2831, 1744, 1678, 1630, 1536, 1435, 1406, 1313, 1202, 1012, 951 cm-1; ”

Author Response

The new typos found by this reviewer have been corrected